# Is Women’s Engagement in Women’s Development Groups Associated with Enhanced Utilization of Maternal and Neonatal Health Services? A Cross-Sectional Study in Ethiopia

**DOI:** 10.3390/ijerph20021351

**Published:** 2023-01-11

**Authors:** Fisseha Ashebir Gebregizabher, Araya Abrha Medhanyie, Afework Mulugeta Bezabih, Lars Åke Persson, Della Berhanu Abegaz

**Affiliations:** 1Tigray Regional Health Bureau, Mekelle P.O. Box 07, Ethiopia; 2School of Public Health, College of Health Sciences, Mekelle University, Mekelle P.O. Box 1871, Ethiopia; 3Department of Disease Control, Faculty of Infectious and Tropical Diseases, London School of Hygiene & Tropical Medicine, London WC1E 7HT, UK; 4Ethiopian Public Health Institute, Addis Ababa P.O. Box 1242, Ethiopia

**Keywords:** antenatal and prenatal care, engagement, maternal and neonatal health, knowledge, postnatal care, women’s development group

## Abstract

Background: In Ethiopia, the Women Development Group program is a community mobilization initiative aimed at enhancing Universal Health Coverage through supporting the primary healthcare services for mothers and newborns. This study aimed to assess the association between engagement in women’s groups and the utilization of maternal and neonatal health services. Method: A cluster-sampled community-based survey was conducted in Oromia, Amhara, Southern Nations, Nationalities and Peoples, and Tigray regions of Ethiopia from mid-December 2018 to mid-February 2019. Descriptive and logistic regression analyses were performed, considering the cluster character of the sample. Results: A total of 6296 women (13 to 49 years) from 181 clusters were interviewed. Of these, 896 women delivered in the 12 months prior to the survey. Only 79 (9%) of these women including Women Development Group leaders reported contact with Women Development Groups in the last 12 months preceding the survey. Women who had educations and greater economic status had more frequent contact with Women Development Group leaders. Women who had contact with Women Development Groups had better knowledge on pregnancy danger signs. Being a Women Development Group leader or having contact with Women Development Groups in the last 12 months were associated with antenatal care utilization (AOR 2.82, 95% CI (1.23, 6.45)) but not with the use of facility delivery and utilization of postnatal care services. Conclusions: There is a need to improve the organization and management of the Women Development Group program as well as a need to strengthen the Women Development Group leaders’ engagement in group activities to promote the utilization of maternal and neonatal health services.

## 1. Introduction

Studies show that community health workers and community volunteers may make significant contributions to a well-functioning primary healthcare system [1,2,3]. In low- and middle-income countries with limited financial resources and professional health taskforce and a high proportion of poor health outcomes, community workers may contribute to making universal health coverage a reality [2,4,5].

Low- and middle-income countries face challenges in training and retaining a sufficient number of health workers [6,7]. As part of task shifting, there has been a move to train lay persons to support and work in the health system [7,8]. There are several names for such workers, including community health volunteers that may be defined as “individuals delivering a health-related service to the community” and “who do not receive a regular salary and/or hold a formal position within the health system” [2,8].

In 2003, the Government of Ethiopia introduced the Health Extension Program [9]. More than 38,000 full-time salaried civil servants, called health extension workers, were deployed to rural parts of the country [10]. In 2011, the government established the volunteer Women’s Development Group (WDG) initiative [10], which is similar to the concept of community health volunteers in other countries [8,11]. Each WDG is composed of 25 to 30 women in one neighborhood and is led by one woman. Each WDG is further divided into a sub-structure of one-to-five networks; one network includes six women of which one is the network leader [11,12].

The roles and practices of community health volunteers vary based on the country’s historical, political, and economic context [2]. In Ethiopia, the WDG initiative is a women-centered approach used to mobilize and engage the leaders’ respective neighbors. It aims to enhance community participation and community ownership of the primary healthcare activities, improving healthcare-seeking behavior and, in particular, the use of maternal and neonatal health services [11,12,13].

The WDG is regarded as a key vehicle to support and promote community ownership of the Health Extension Program to achieve Ethiopia’s Health Sector Transformation Plan as well as universal health coverage. This plan is aligned with the maternal and neonatal health targets set by the Sustainable Development Goals [9,14].

A global health priority has contributed and shown to be vital towards achieving universal health coverage, mainly regarding reproductive, maternal, neonatal and child health services as part of the Sustainable Development Goals [5]. Although the proportion of mothers dying per 100,000 live births declined from 1400 in 1990 to 401 in 2017 (World Health Statistics, 2021), the ratio is still high in Ethiopia [15]. Similarly, the infant mortality rate has declined from 59.6 per 1000 live births in 2009 to 36.5 per 1000 live births in 2019 [16]. Hence, Ethiopia has come far in regard to a position to meet universal health coverage by 2030. Women’s group activities may create trust in the health system and increase the utilization of maternal, neonatal, and child health services [17,18]. Despite some improvements in antenatal and perinatal care and facility delivery, postnatal care utilization coverage remains low [16]. Women with low or no education, living far from health facilities, and without available transport have difficulties in using health services. In addition, there are sometimes socio-cultural barriers and low awareness of danger signs for women in the postnatal period [8,19]. WDG leaders are expected to counsel and mobilize women to use maternal health services [4,12]. They are also expected to identify pregnant women, organize pregnant women’s forums, visit newborns, refer sick children to the health posts, and counsel families to follow-up on referrals [14,20,21]. Some studies have shown that women exposed to WDGs were more likely to utilize maternal, neonatal, and child health services, potentially contributing to the realization of universal health coverage by 2030 [13,16,22] and ultimately to a reduction in maternal and child mortality [6,23].

Studies have shown WDG leaders’ knowledge, functionality, and performance to be insufficient for such purposes [10,12,24]. They had a heavy workload [25], faced a lack of trust from the community, and had insufficient supervision [8,26]. In addition, their lack of knowledge on maternal, neonatal, and child health reportedly affected their potential to increase the utilization of maternal and child health services in their respective communities [8,12,27]. These issues have affected the frequency and quality of the WDG leaders’ contact with their members [12,27]. As a result, the WDG leaders’ performance has been reported to be low [8,28].

Worldwide, there is an increasing interest in community health volunteers’ performance. Although studies have shown that they are able to deliver certain preventive, promotive, and even curative services, a systematic review revealed that there were still gaps in the evidence as to whether community health volunteers could contribute to improvements in population health [2]. In order to improve the utilization of maternal and child health services, the Ethiopian Government introduced the Optimizing Health Extension Program (OHEP) intervention in four regions that aimed to increase the utilization of primary maternal, neonatal, and child health services. This study is a secondary analysis of data from the endline survey which aimed to assess whether women’s engagement in women’s group activities was associated with the utilization of maternal and neonatal health services.

## 2. Methods

### 2.1. Study Design and Setting

The Optimizing Health Extension Program intervention was implemented in 26 districts. The evaluation was conducted in these 26 intervention districts and 26 comparison districts that were selected to be similar to the intervention districts based on demographic and health service characteristics [17]. A baseline study was conducted before the intervention (mid-December 2016 to mid-February 2017) and an endline study was conducted two years after the baseline survey (mid-December 2018 to mid-February 2019). Our secondary analysis was unrelated to the original evaluation study of the OHEP intervention. Therefore, we used data from the endline survey and combined the data from the original intervention and comparison areas [17].

The study design was a cluster-sampled cross-sectional survey and was carried out in Oromia, Amhara, Southern Nations, Nationalities and Peoples, and Tigray regions of Ethiopia. Figure 1 shows a map of the study districts.

### 2.2. Data Source

A two-staged stratified cluster sampling method was used to select the study subjects. The 2007 Ethiopian Housing and Population Census data were used to identify and list enumeration areas in the 52 study districts. The plan was to select 200 enumeration areas proportional to the size of the districts. However, by the endline survey, nineteen clusters were excluded due to civil unrest, and 181 enumeration areas were finally considered for the study. In each enumeration area, 66 households were selected and all women who were eligible for interview (13 to 49 years old) and available on the day of the survey were invited to participate in the study.

A total of 120 data collectors were recruited by the Ethiopian Public Health Institute and had, as a minimum, completed their first degree in health sciences. The data collectors were trained for two weeks, including in study procedures, questionnaires, interview and data collection techniques, quality assurance procedures, and study ethics. A field manual was prepared and used in the training. Regional coordinators and team leaders were assigned to supervise the interviews. The interviews were conducted face-to-face. On average, the interviews took thirty minutes. One interviewer interviewed on average six households per day.

A questionnaire was prepared in English and translated into the local languages Amharic, Oromiffa, and Tigrigna, and after that back-translated into English. The questionnaire was based on earlier validated survey tools, including the Ethiopia Demographic and Health Survey tools which were used for the household survey and other tested tools, which were used for the evaluation of the integrated Community Case Management of Childhood Illnesses and Community Based Newborn Care programs [21,29].

The household survey comprised of two sections. The first was a household overview administered to the head of the household to collect information on age and sex of all current residents, the identification of the primary caregiver to any child under five years of age, characteristics of the house and its assets, access to healthcare, and location of each household based on geographical positioning system (GPS) assessment. The second was the women of reproductive age (13 to 49 years) section capturing information on the healthcare available to them, and their recent contact with WDG leaders and health providers. The decision to include a 1-year recall period was primarily related to the original OHEP evaluation to reduce recall bias for details. In addition, the majority of the respondents were illiterate; we decided to consider women who had a last birth history for children born in the past 12 months preceding the survey to minimize recall bias. Furthermore, the interview included care seeking and utilization of health services during pregnancy, delivery, and postpartum for the first 28 days after delivery periods [17,30].

### 2.3. Measurements

#### Outcome Variables

The analysis considered three maternal and neonatal health dependent variables: antenatal care service utilization, use of facility delivery, and utilization of postnatal care services. Antenatal or prenatal care service utilization was the indicator of access and use of healthcare services during pregnancy. At least one antenatal care visit defined the proportion of pregnant women who received at least one example of antenatal care for a pregnancy in the 12 months preceding the survey. Use of facility delivery captured the proportion of live births (13 to 49 year old women) at a health facility that had been assisted by health professionals within the last 12 months preceding the survey. The utilization of postnatal care indicator defined the proportion of women who received postnatal care in the first month after delivery for a pregnancy that occurred in the 12 months preceding the survey. Utilization of postnatal care during the first month after delivery assumed that the use was for both mothers and newborns. This analysis included the WDG contact in the 12 months prior to the survey as the main contact outcome variable. Assessment of women’s contact with WDG leaders was categorized as follows: women who had WDG contact but who were not WDG leaders; women who were WDG leaders; and women who neither were WDG leaders nor had WDG contact. Additional information was also collected on women’s socio-economic characteristics, including their years of education, religion, marital status, birth order, and household wealth. Education was categorized into no education and educated. Educated refers to those who had at least one year of formal education or more. Religions included Christian Orthodox, Muslims, and Protestant. Birth order refers to the order a child is born in within their family. Birth order was classified into one child, two or three children, and four or more children. The household wealth index was based on durable assets, household building materials, utilities, and animals owned. The continuous variable produced by the first principal component was divided into five equally (20%) sized groups (quintiles) of households from quintile 1 (poorest) to quintile 5 (wealthier).

### 2.4. Data Analysis

Data were collected on personal tablet computers, and the Census and Survey Processing System (CSPro) was used. The collected data were regularly sent to the central server at the Ethiopian Public Health Institute. The server was password protected, and access to the data was limited to the study team. Data were cleaned and prepared for analysis.

The quality of the information collected was ensured by using validated and pretested forms, a system of field supervision, and careful data quality control and management that included daily checks on completeness and consistency. A detailed survey manual with extensive standard operating procedures was prepared and used in training, piloting, and fieldwork.

A descriptive analysis, including frequencies and percentages, was performed. Background factors, including contact with a WDG leader or being a WDG leader in the last 12 months, were cross-tabulated with antenatal care service utilization, use of facility delivery, and utilization of postnatal care. Fisher’s exact test was performed to determine the significant association for variables with small values. A multivariate analysis was thereafter conducted to assess whether there was an association between having a WDG contact or being a WDG leader and antenatal care service utilization, use of facility delivery, and utilization of postnatal care services. We controlled for factors that were significantly associated in the bivariate analysis. The “svyset” and “svy” prefixes were used to adjust for the cluster sample design to account for a clustering effect. The STATA statistical software version 14.0 (StataCorp LLC., College Station, TE, USA) was used for all analyses.

Ethical review: Ethical approval was obtained from the Ethiopian Public Health Institute (protocol number SERO-012-8-2016), London School of Hygiene and Tropical Medicine (protocol number 11,235), and the Institutional Review Board (IRB) of Mekelle University, College of Health Sciences (protocol number 1433/2018). Support letters were also obtained from the Regional Health Bureaus in Amhara, Oromia, Southern Nations and Nationalities Peoples, and Tigray. Informed consent was obtained from all study participants.

## 3. Results

### 3.1. Background Characteristics of Study Participants

A total of 6296 women from 181 clusters were interviewed in the endline survey. Of these, 896 women had delivered in the 12 months prior to the survey. The socio-demographic characteristics of these women are shown in Table 1. The mean age was 28 years, and 460 (51%) of the women had no education. Five hundred eighty-eight (66%) were Christians and 33% were Muslim. Almost all (95%) were married. Six hundred eighty-five (77%) had received at least one example of antenatal care, four hundred sixty-four (52%) had delivered in a facility, and one hundred fifty-five (17%) had utilized a postnatal care visit in the first month after delivery. Only 20 (2%) of the women had had contact with WDG leaders in the last 12 months and 59 (7%) were leaders of a WDG (Table 1).

A greater proportion of women who had educations and greater economic status had contact with WDG leaders.

### 3.2. Knowledge of Pregnancy Danger Signs

Women who had had contact with WDG leaders or were the WDG leaders themselves had better knowledge on nearly all pregnancy danger signs. Women who had WDG contact but were not WDG leaders had even better knowledge than WDG leaders on some of the pregnancy danger signs (Figure 2).

### 3.3. Factors Associated with Antenatal Care, Facility Delivery, and Postnatal Care

Women who were either visited by a WDG leader in the last 12 months or were WDG leaders themselves had, to a larger extent, utilized antenatal care, compared to those with no WDG contact (Table 2). This engagement with WDGs was also significantly associated with the use of facility delivery, compared to those not exposed to WDGs. Engagement in WDGs was not significantly associated with the utilization of postnatal care in the first month after delivery.

Education, economic status, and religion were significantly associated with antenatal care utilization and use of facility delivery (Table 2). Birth order was also significantly associated with use of facility delivery. Education and economic status were also significantly associated with utilization of postnatal care.

Factors associated with antenatal care utilization, use of facility delivery, and utilization of postnatal care were included in a multivariable logistic regression. The WDG leaders or those who had been in contact with a WDG leader in the last 12 months were more likely to have utilized an antenatal care service (AOR 2.82, 95% CI (1.23, 6.45), *p*-value = 0.01) (Table 3). There was no significant association between WDG contact and the use of facility delivery or the utilization of postnatal care services (Table 3).

## 4. Discussion

Overall, very few women had contact with WDG leaders. Women who had educations or were from wealthier households had more contact with WDG leaders or were leaders themselves. Women who had contact with WDG leaders or were leaders had better knowledge of pregnancy danger signs. WDG engagement (contact or being the leader) was significantly associated with antenatal care service utilization, but not with the use of facility delivery or utilization of postnatal care services.

Prior to 2016, WDG leaders played a pivotal role in the health status of women due to close and continuous contact with the community, in particular with women [6,10,31]. The WDG network leaders were expected to meet with their members every other day [12]. The present study revealed that very few women had contact with WDG leaders during the last 12 months preceding the survey. This is the first study reporting that women had very infrequent contact with WDGs, although these findings are in line with some previous studies conducted in Ethiopia [8,9,22]. The potential reasons for the low contact could be that WDG leaders have a high workload. They could also be fatigued due to their other duties and their many years of service. Perhaps their contact has also decreased with an increasing lack of trust from the community. These issues are believed to have contributed to a poorly functioning WDG program.

At the community level, the implementation of the WDG program has been unclear with large variations [2,32]. Women who had some education or better wealth had better contact with WDGs. Education and wealth are important enabling factors for the use of all primary healthcare services [32,33]. Hence, level of education of women can be used as a core criterion for the selection of health cadres in the study communities. Furthermore, women who had fewer children were more likely to receive the use of facility delivery [34]. These findings might be owed to women’s education status, better wealth, and experience of antenatal care which enable improved healthcare seeking.

Women who had contact with WDG leaders or were WDG leaders themselves had more knowledge on nearly all pregnancy danger signs. Studies have shown that knowledge of pregnancy danger signs is of great importance to improve women’s awareness and use of services and ultimately improve their own and their neonates’ health [12,18,35]. Although women who had contact with WDG leaders or were WDG leaders themselves had more knowledge on pregnancy danger signs compared to women who had no WDG contact, their knowledge was still sub-optimal. This is also similar to the finding of a qualitative study conducted in the same project [8]. This could be due to a lack of support from the health extension workers in the primary healthcare system. Improving the knowledge of WDG leaders could lead to better engagement with their members. This could enhance their community health outreach and timely home visits to counsel, arrange community referrals, and ensure reduced delays in seeking care [36].

More than three quarters of the women had received at least one example of antenatal care and half had delivered in a facility, but only less than one fifth had received postnatal care in the first month after delivery. The 2019 Ethiopia Mini-Demographic and Health Survey (EMDHS) and the EDHS 2016 datasets showed consistent findings with this study [16,19]. Women who had contact with WDG leaders or were WDG leaders more frequently utilized at least one antenatal care service [4,6,22]. There were no significant associations between WDG contact and the use of facility delivery and utilization of postnatal care. Despite increasing trends of antenatal care and facility delivery, the absolute level and gap with regard to the target levels expressed in the Ethiopia Health Transformation Plan 2020 remain remarkably large [4,37,38]. These findings may be caused by the low attention given to community health engagement and inadequate investment in the WDG program. This implies that an overall poor linkage between the community and the health system persists, leading to weak organization and management of the WDG program [8].

Although the sample was selected to represent districts in the four regions of Ethiopia, it is likely that findings are typical for most rural parts of these regions. The levels of maternal and neonatal health service utilization are similar to the corresponding Ethiopian Demographic and Health Survey results. The small number of women with WDG contact limits some of the analyses. This was managed by combining the categories “contact with” and “being a WDG leader”. Due to the cross-sectional nature of the study design, causal inferences cannot be made.

## 5. Conclusions

We have shown that engagement in women’s development groups may potentially enhance women’s attendance at antenatal care appointments. However, the study findings point at the poor functionality of WDGs. The Women Development Group program needs to be revitalized in its organization and management. It could also benefit from task shifting for selected health services and improved criteria for the recruitment of leaders and support from the primary healthcare system.

## Figures and Tables

**Figure 1 ijerph-20-01351-f001:**
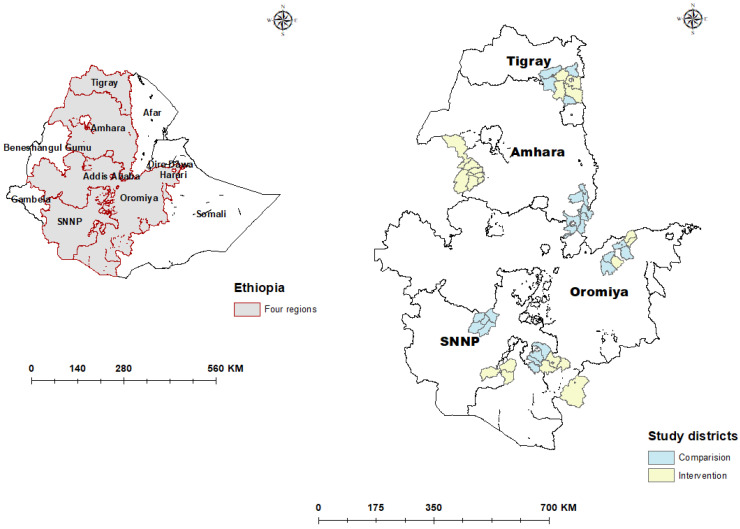
Map of Ethiopia showing all regions (**left**) and the intervention and comparison districts within the four study regions (**right**), 2018/2019 [17].

**Figure 2 ijerph-20-01351-f002:**
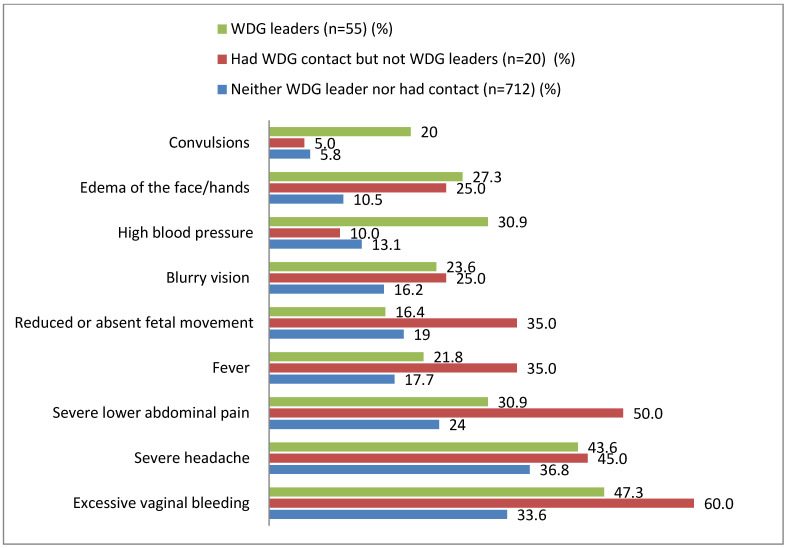
Knowledge of women who delivered last year on pregnancy danger signs by level of contact to the WDG, Ethiopia, 2018/19. WDG = Women’s development group.

**Table 1 ijerph-20-01351-t001:** Characteristics of women who delivered in the 12 months prior to the study in Ethiopia, 2018/2019.

Characteristics	N = 896 (%)
Education level of women	
No education	460 (51.3)
Educated	436 (48.7)
Religion	
Orthodox Christians	450 (50.2)
Protestant Christians	138 (15.4)
Muslim	299 (33.4)
Other	9 (1.0)
Marital status (*n* = 891) ^a^	
Non-married	13 (1.5)
Married/In a union	845 (94.8)
Divorced/Widowed	33 (3.7)
Birth order (*n* = 892) ^b^	
One child	129 (14.5)
Two or three children	306 (34.3)
Four or more children	457 (51.2)
Received antenatal care	685 (76.5)
Had a facility delivery	464 (51.8)
Received postnatal care in the first month after delivery	155 (17.3)
WDG contact (one year prior to the survey)	
Neither WDG leader nor had contact	817 (91.2)
Had WDG contact but not a WDG leader or was a WDG leader	79 (8.8)

^a^ Missing data on 5 women; ^b^ missing data on 4 women.

**Table 2 ijerph-20-01351-t002:** WDG contact and distribution of socio-economic characteristics of women across utilization of maternal and neonatal health services in Ethiopia, 2018/19.

Characteristics	Continuum of Maternal and Neonatal Health Services								
	Antenatal Care Service Utilization			Use of Facility Delivery			Utilization of Postnatal Care		
	Yes (%) (*n* = 685)	No (%) (*n* = 211)	*p*-Value b	Yes (%) (*n* = 464)	No (%) (*n* = 432)	*p*-Value b	Yes (%) (*n* = 155)	No (%) (*n* = 741)	*p*-Value b
WDG ^a^ contact			0.001 *			0.013 *			0.317
Neither WDG leader nor had contact	613 (75.0)	204 (25.0)		413 (50.6)	404 (49.4)		138 (16.9)	679 (83.1)	
Had WDG contact but not WDG leader, or WDG leader	72 (91.1)	7 (8.9)		51 (64.6)	28 (35.4)		17 (21.5)	62 (78.5)	
Education level of women			0.006 *			0.017 *			0.022 *
No education	333 (72.4)	127 (27.6)		217 (47.2)	243 (52.8)		65 (14.1)	395 (85.9)	
Educated	352 (80.7)	84 (19.3)		247 (56.7)	189 (43.3)		90 (20.6)	346 (79.4)	
Religion **			<0.000 *			<0.000 *			0.581
Orthodox	388 (86.2)	62 (13.8)		316 (70.2)	134 (29.8)		86 (19.1)	364 (80.9)	
Protestant	108 (78.3)	30 (21.7)		49 (35.5)	89 (64.5)		22 (15.9)	116 (84.1)	
Muslim	184 (61.5)	115 (38.5)		96 (32.1)	203 (67.9)		46 (15.4)	253 (84.6)	
Other	5 (55.6)	4 (44.4)		3 (33.3)	6 (66.7)		1 (11.1)	8 (88.9)	
Economic status			<0.001 *			<0.001 *			<0.001 *
Wealth quintile one	148 (62.5)	89 (37.6)		70 (29.5)	167 (70.5)		22 (9.3)	215 (90.7)	
Wealth quintile two	138 (78.9)	37 (21.1)		81 (46.3)	94 (53.7)		20 (11.4)	155 (88.6)	
Wealth quintile three	131 (79.9)	33 (20.1)		89 (54.3)	75 (45.7)		31 (18.9)	133 (81.1)	
Wealth quintile four	127 (76.5)	39 (23.5)		97 (58.4)	69 (41.6)		33 (19.9)	133 (80.1)	
Wealth quintile five	141 (91.6)	13 (8.4)		127 (82.5)	27 (17.5)		49 (31.8)	105 (68.2)	
Marital status **	*n* = 681	*n* = 210	0.741	*n* = 462	*n* = 429	0.215	*n* = 154	*n* = 737	0.392
Non-married	9 (69.2)	4 (30.8)		6 (46.2)	7 (53.8)		3 (23.1)	10 (76.9)	
Married/union	647 (76.6)	198 (23.4)		434 (51.4)	411 (48.6)		148 (17.5)	697 (82.5)	
Divorced/widowed	25 (75.8)	8 (24.2)		22 (66.7)	11 (33.3)		3 (9.1)	30 (90.9)	
Birth order	*n* = 684	*n* = 208	0.418	*n* = 463	*n* = 429	<0.000 *	*n* = 155	*n* = 737	0.077
One child	105 (81.4)	24 (18.6)		91 (70.4)	38 (29.6)		31 (24.0)	98 (76.0)	
Two or three children	233 (75.2)	73 (24.8)		161 (52.6)	145 (47.4)		53 (17.3)	253 (82.7)	
Four or more children	346 (75.7)	111 (24.3)		211 (46.2)	246 (53.8)		71 (15.5)	386 (84.5)	

^a^ WDG = Women development group; ^b^
*p*-value for chi-square; adjusted for clustering; * *p*-value < 0.05; ** Fisher’s exact test.

**Table 3 ijerph-20-01351-t003:** Association of WDG contact with utilization of maternal and neonatal health services in Ethiopia, 2018/19.

WDG ^a^ Contact	Utilization of Maternal and Neonatal Health Services								
	Antenatal Care Service Utilization			Use of Facility Delivery			Utilization of Postnatal Care		
	OR (95% CI)	AOR (95% CI) ^b^	*p*-Value	OR (95% CI)	AOR (95% CI) ^c^	*p*-Value	OR (95% CI)	AOR (95% CI) ^d^	*p*-Value
Neither WDG ^a^ leader nor had contact	1.00	1.00		1.00	1.00		1.00	1.00	
WDG ^a^ leader or had WDG contact but not WDG leader	3.34 (1.56, 7.51)	2.82 (1.23, 6.45)	0.014	1.78 (1.12, 2.82)	1.34 (0.77, 2.32)	0.301	1.37 (0.75, 2.44)	1.24 (0.64, 2.39)	0.510

^a^ WDG = Women’s development group; ^b^ adjusted for clustering, education level, religion, and economic status; ^c^ adjusted for clustering, education level, religion, economic status, and birth order; ^d^ adjusted for clustering, education level, economic status, and birth order.

## Data Availability

The dataset used in this study can be made available from the Ethiopia Public Health Institute via Martha Zeweldemariam (E-mail: Martha.zeweldemariam@lshtm.ac.uk), after contacting the corresponding author.

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
