# Peer review of "Is Women’s Engagement in Women’s Development Groups Associated with Enhanced Utilization of Maternal and Neonatal Health Services? A Cross-Sectional Study in Ethiopia"

_ijerph, 2023, doi:10.3390/ijerph20021351_

Round 1
Reviewer 1 Report
Dear Authors
The article presented " Is women's engagement in Women's Development Groups associated with enhanced utilization of maternal and neonatal health services? A cross-sectional study", is supported by interesting objectives with interest for the practice. Some recent studies show that Community health workers (CHWs) have been identified as good strategies for reaching these communities because of their geographical and social proximity to community residents and the cost-effectiveness of engaging them in the service delivery of health. In low- and middle-income countries with few resources of physicians and midwives, low national health spending and a high proportion of poor health outcomes, community workers are considered key players in making universal health coverage a reality.
The study planning was done correctly, although the methodology is not described in sufficient detail to allow an understanding of the results obtained.
As long as I can evaluate myself as a non-native English speaker, the language is adequate and correct. Some aspects of linguistic and grammatical correction can be improved.
Some details might be improved to increase the clarity of the manuscript suggested below.
Abstract
Line 20 – "…were interviewed", consider replacing for: were questioned;
Line 26 – "Mothers who had had contact with WDG leaders…" consider replacing for: Women who had contact with WDG leaders…
Line 29 to 30- "Training and support to Women's Development Group leaders need tailored interventions". The study does not allow for these conclusions, these are inferential from the authors, perhaps because they have other knowledge of the broader study.
Keywords: Authors should consider reviewing the study's keywords, for example, I suggest removing-Contact; Engagement- and including Prenatal care, Postnatal care; maternal and newborn health;
Introduction
I would recommend that they could better contextualize the difficulties in accessing prenatal and postnatal health care for women in these communities and what health outcomes have already been identified from this type of community support. How prepared are these groups of women who provide community support?
Line 78 to 79 – "For the evaluation of this initiative, surveys were performed [11]". The suggested removing this information does not make sense in the introduction.
Line 79 to 82 - I recommend revising the objective in the abstract and introduction to match. In the introduction, the objective does not seem well formulated, as I think that this study did not evaluate assistance, but only the demand for health services.
Methodology
Although this study is part of a broader investigation, it would be important to clarify some of the methodological aspects for a better understanding of the study carried out.
Type of study, Criteria for inclusion of participants, description of instruments used.
Line 90 to 91 - "We used the endline data and have combined data from both intervention and comparison areas." - I suggest that this idea be clarified because it is not clear what they mean.
Line 129 – "The score for "yes" and "no" were 1 and 0 respectively" - This information appears out of context and it is not clear what it concerns.
Line 143 to 148 - This information is out of place and belongs to the previous point where the variables are presented, not to the statistical analysis.
Line 162 –"… bivariate analysis and svyset was used before…" - What it means svyet ??
Results
Line 173 to 178 – I suggest that you can put the percentages because it is more informative.
Line 180 - I recommend that the analysis of results begin with the sociodemographic characterization, so this information should come right at the beginning.
Line 181 - In table 1, lines 14, 15 and 16 remove the percentage symbol.
Line 187 – I suggest replacing Mothers, for women. It would be important to understand how this knowledge about complications in pregnancy was evaluated because the methodology is not clear.
Line 212 -Table 3. - If the confidence interval includes a value of 1, this implies that there is no difference between the groups studied, however, as they used binary logistic regression, I think that significance values (p-value) should be included.
Line 194 to 202 - I would suggest that this chapter of the results be clarified what was intended to be evaluated (use, utilization, visited, assistance), as the term is used in an undifferentiated way and conceptually has different meanings that should be clarified in the methodology, given the importance of this variable in the study.
Line 200 - It is not clear what the meaning of the association studied is.
Discussion
I recommend that they be able to include what brings the study new to existing knowledge and that proposal for improving care be discussed based on the results.
The discussion is redundant and I think it would be important for the authors to highlight the real contribution of these results to the planning of maternal and obstetric health care in these rural communities.
Include study limitations including the small number of participants who had contact with Women's Development Groups. "Of these, 896 women, …. only 20 (2%) of these women reported contact with Women's Development Group leaders in the last 12 months preceding the survey and another 59 (7%) were WDG leaders".
Conclusions
The conclusions are quite generic and should be more specific considering the results of the study. It is crucial to analyse and reflect on health policies, skills training for caregivers and community outreach measures.
Very successful for your research.
Author Response
Response to Reviewer 1
Comments Comments and Suggestions
for Authors Dear Authors The article presented " Is women's engagement in Women's Development Groups associated with enhanced utilization of maternal and neonatal health services? A cross-sectional study", is supported by interesting objectives with interest for the practice. Some recent studies show that Community health workers (CHWs) have been identified as good strategies for reaching these communities because of their geographical and social proximity to community residents and the cost-effectiveness of engaging them in the service delivery of health. In low- and middle-income countries with few resources of physicians and midwives, low national health spending and a high proportion of poor health outcomes, community workers are considered key players in making universal health coverage a reality. Response: We thank you for your guidance. We have added your pionts with appropriate references (Introduction section, on page 2, in lines 41 to 45). The study planning was done correctly, although the methodology is not described in sufficient detail to allow an understanding of the results obtained. Response: Thank you. We have expanded the methods section for improved clarity As long as I can evaluate myself as a non-native English speaker, the language is adequate and correct. Some aspects of linguistic and grammatical correction can be improved. Response: Thank you. We have made some further language corrections. Some details might be improved to increase the clarity of the manuscript suggested below. Abstract Line 20 – "…were interviewed", consider replacing for: were questioned; Response: Thank you for the suggestion. Since the data collection method was an interview, we believe keeping “interviewed” is most appropriate (Abstract section, page 1, line 19). Line 26 – "Mothers who had had contact with WDG leaders…" consider replacing for: Women who had contact with WDG leaders… Response: Thank you. Corrected as suggested (Abstract section, page 1, line 23; and other sections) Line 29 to 30- "Training and support to Women's Development Group leaders need tailored interventions". The study does not allow for these conclusions, these are inferential from the authors, perhaps because they have other knowledge of the broader study. Response: Thank you. We have deleted this sentence (Abstract section, on page, line 29). Keywords: Authors should consider reviewing the study's keywords, for example, I suggest removing-Contact; Engagement- and including Prenatal care, Postnatal care; maternal and newborn health; Response: Thank you. We have changed keywords accordingly (Keywords, page 2, lines 37 and 38). Introduction I would recommend that they could better contextualize the difficulties in accessing prenatal and postnatal health care for women in these communities and what health outcomes have already been identified from this type of community support. How prepared are these groups of women who provide community support? Response: Thank you. We have edited the introduction to focus on community support (Introduction section, page 3, lines 77 to 84). Line 78 to 79 – "For the evaluation of this initiative, surveys were performed [11]". The suggested removing this information does not make sense in the introduction. Response: It is a well-accepted comment. We have deleted this sentence (Introducation section, on page 4, line 103). Line 79 to 82 - I recommend revising the objective in the abstract and introduction to match. In the introduction, the objective does not seem well formulated, as I think that this study did not evaluate assistance, but only the demand for health services. Response: We thank you for this comment. We have revised the objective in the Introduction that now matches the corresponding sentence in Abstract (Introduction section, on page 4, lines 104 to 106). Methodology Although this study is part of a broader investigation, it would be important to clarify some of the methodological aspects for a better understanding of the study carried out. Type of study, Criteria for inclusion of participants, description of instruments used Response: We thank you for this comment. We have mentioned the study design in page 5, line 139. The inclusion criteria were all women who were eligible for interview (13 to 49 years old) and available on the day of the survey were invited to participate in the study (Methods section, page 6, lines 155 and 156). We have also described the “questionnaire/ instruments” how we used (Methods section, pages 6 and 7, lines 164 to 183). Line 90 to 91 - "We used the endline data and have combined data from both intervention and comparison areas." - I suggest that this idea be clarified because it is not clear what they mean. Response: Thank you. Since the aim of this analysis was unrelated to the original evaluation study of the OHEP intervention, we used data from intervention as well as comparison areas (Methods section, page 5, lines 136 to 138). (See more; Berhanu D, Okwaraji YB, Defar A, et al. Does a complex intervention targeting communities, health facilities and district health managers increase the utilisation of community based child health services? A before and after study in intervention and comparison areas of Ethiopia. BMJ Open 2020;10:e040868. doi:10.1136/ bmjopen-2020-040868; and Berhanu D, Okwaraji YB, Belayneh AB, Lemango ET, Agonafer N, Birhanu BG, et al. Protocol for the evaluation of a complex intervention aiming at increased utilisation of primary child health services in Ethiopia: a before and after study in intervention and comparison areas. BMC Health Serv Res. BioMed Central 20:339. doi.org/10.1186/s12913-020-05151-3). Line 129 – "The score for "yes" and "no" were 1 and 0 respectively" - This information appears out of context and it is not clear what it concerns. Response: Thank you. We have deleted this sentence, as suggested (Methods section, on page 8, lines 215). Line 143 to 148 - This information is out of place and belongs to the previous point where the variables are presented, not to the statistical analysis. Response: Thank you for pointing this out. We have moved this information to methods section, on page 7 and 8, lines 198 to 215. Line 162 –"… bivariate analysis and svyset was used before…" - What it means svyet ?? Response: We thank you for this comment. We thank you for this comment. The “svyset” and “svy” are the survey prefix command for survey set. The “svyset” and “svy”: prefix can be used with many statistical commands to allow for survey sample design; while considering the clustering effect (Methods section, on page 9, in lines 281 and 282). Results Line 173 to 178 – I suggest that you can put the percentages because it is more informative. Response: There are well-accepted comments. The change has been made according to your comment ((Results section, page 10, lines 296 to 298). Line 180 - I recommend that the analysis of results begin with the socio-demographic characterization, so this information should come right at the beginning. Response: We agree with you; we have changed this (Results section, on page 10, line 295). Line 181 - In table 1, lines 14, 15 and 16 remove the percentage symbol. Response: Response: Thank you, corrected. (Results section, Table 1, page 10, line 336). Line 187 – I suggest replacing Mothers, for women. It would be important to understand how this knowledge about complications in pregnancy was evaluated because the methodology is not clear. Response: Thank you. We have made this change throughout the manuscript. We have also edited the Methods section for clarity (Results section, on page 11, line 361 and in figure 2, line 366). Line 194 to 202 - I would suggest that this chapter of the results be clarified what was intended to be evaluated (use, utilization, visited, assistance), as the term is used in an undifferentiated way and conceptually has different meanings that should be clarified in the methodology, given the importance of this variable in the study. Response: We thank you for this comment. We have edited the text regarding outcome variables for clarity as follows: the analyses consider three maternal and neonatal health variables: antenatal care service utilization, use of facility delivery and utilization of postnatal care (Methods section, on page 7, lines 186 and 187). Line 200 - It is not clear what the meaning of the association studied is. Response: The text has been edited for clarity (Results section, on pages 11 and 12, line 372 to 386). Line 212 -Table 3. - If the confidence interval includes a value of 1, this implies that there is no difference between the groups studied; however, as they used binary logistic regression, I think that significance values (p-value) should be included. Response Thank you. We have edited as follows: A multivariate analysis was thereafter conducted to check if there was an association between having a WDG contact or being a WDG leader and antenatal care service utilization, use of facility delivery and utilization of postnatal care. We controlled for factors that were significantly associated in the bivariate analysis. These factors were included in the final multi-variable logistic regression. (Results section, on page 14 line 497, on table 3, line 500). Discussion I recommend that they be able to include what brings the study new to existing knowledge and that proposal for improving care be discussed based on the results. The discussion is redundant and I think it would be important for the authors to highlight the real contribution of these results to the planning of maternal and obstetric health care in these rural communities. Response: Thank you very much for this comment. We have edited the discussion to ensure that it reflects the contribution of community health volunteers in the results, as well as perspectives of this study. Include study limitations including the small number of participants who had contact with Women's Development Groups. "Of these, 896 women, …. only 20 (2%) of these women reported contact with Women's Development Group leaders in the last 12 months preceding the survey and another 59 (7%) were WDG leaders". Response: Thank you. We have added this limitation in the discussion section on pages 16 & 17, lines 608 to 617. Conclusions The conclusions are quite generic and should be more specific considering the results of the study. It is crucial to analyse and reflect on health policies, skills training for caregivers and community outreach measures. Response: We thank you. We have re-written the conclusion as you suggested in the conclusion section on pages 17, lines 620 to 624.

Reviewer 2 Report
The study is very interesting, lovely and essential to put the spotlight on the women of that country and the work done with them. Nevertheless, the data they present are very old, before COVID. The world has changed a lot after the pandemic. It would be necessary to know what has happened in the 3 years since. The baseline data is 6 years old. Currently the obsolescence of the science is dated at 7 years.
Data Source:
There is a lot of data needed to really understand the data collection procedure: how many interviewers conducted interviews?were they paid per interview conducted?did they have to conduct a certain number of interviews, over a certain period of time?how long did they conduct interviews? By what method (telephone, face-to-face, etc.?), did anyone supervise how they conducted the interview?how many items were in the interview?how much time did they spend interviewing on average?how many refused to interview? Etc.
3. Results
3.1. Background characteristics of study participants
If most of the interviewees have had 3 or more children, why haven't they modified this variable to have a more adjusted data?
the discussion is very poor, more involvement is needed with the results found. what are the proposals to improve women's attendance?what are the future perspectives?what has been learned and what can be put into practice? a much more serious discussion of the data found is needed.
Author Response
Response to Reviewer 2 Comments
Comments and Suggestions for Authors
- The study is very interesting, lovely and essential to put the spotlight on the women of that country and the work done with them. Nevertheless, the data they present are very old, before COVID. The world has changed a lot after the pandemic. It would be necessary to know what has happened in the 3 years since. The baseline data is 6 years old. Currently the obsolescence of the science is dated at 7 years.
Response: Thank you for your encouraging words and constructive feedback on the overall manuscript. The data used in this study were collected in early 2019. Baseline data was not used in this article. Reporting was delayed by the civil war in our region Tigray-Ethiopia’s with lack of internet access. Currently, we have got limitted internet access and have been able to submit our article.
- Data Source:
There is a lot of data needed to really understand the data collection procedure: did anyone supervise how they conducted the interview? how many interviewers conducted interviews? did they have to conduct a certain number of interviews, over a certain period of time? how long did they conduct interviews? By what method (telephone, face-to-face, etc.?), how many items were in the interview? Were they paid per interview conducted? how much time did they spend interviewing on average? how many refused to interview?
Response: Thank you. We have expanded the description in Methods section.
For more information: A field manual was prepared and used in the training. Regional coordinators and team leaders were assigned to supervise the interviews. A total of 120 data collectors were also recruited and trained for two weeks. The interviews were conducted face-to-face. One interviewer interviewed on average six households per day. The data collectors took two months from Mid-December 2018 to Mid-February 2019. In addition, the endline survey items were included seven items’ interviews. There was not paid for participants. On average, the interviews took thirty minutes. Each person was given a full explanation about the survey and each person was free to agree or not to agree to be interviewed. However, we would like everyone to agree to be interviewed because their answer is important to us, and their answers must be treated as confidential. There was not faced refusal to interview, remain absent from their nearest home. For this we did a replacement mechanism. In communities, we would like to interview Women Development Group leaders and women who were aged 13 to 49 years who support maternal and newborn health about the work they do in the community (Methods section, page 6, lines 157 to 163).
(See more, the protocol paper and previous publication are important for further details on the methods section Berhanu D, Okwaraji YB, Belayneh AB, Lemango ET, Agonafer N, Birhanu BG, et al. Protocol for the evaluation of a complex intervention aiming at increased utilisation of primary child health services in Ethiopia: a before and after study in intervention and comparison areas. BMC Health Serv Res. BioMed Central 20:339. doi.org/10.1186/s12913-020-05151-3); AND Berhanu D, Okwaraji YB, Defar A, et al. Does a complex intervention targeting communities, health facilities and district health managers increase the utilisation of community based child health services? A before and after study in intervention and comparison areas of Ethiopia. BMJ Open 2020;10:e040868. doi:10.1136/ bmjopen-2020-040868.)
- Results
3.1. Background characteristics of study participants
If most of the interviewees have had 3 or more children, why haven't they modified this variable to have a more adjusted data?
Response: Thank you. In order to reduce recall bias, the study focuses women who had given birth in the last 12 months (Methods section, on page 7, lines 180 and 181).
- The discussion is very poor, more involvement is needed with the results found. what are the proposals to improve women's attendance? what are the future perspectives? what has been learned and what can be put into practice? a much more serious discussion of the data found is needed.
Response: Thank you. We have edited the discussion to ensure that it reflects the results regarding the potential contribution of community health volunteers, as well as perspectives of this study in improving the utilization of maternal and neonatal health services.
Reviewer 3 Report
Peer review:
IJERPH
Is women's engagement in Women’s Development Groups associated with enhanced utilization of maternal and neonatal health services? A cross-sectional study
General comments
=============
The manuscript described a cross-sectional study to evaluate the association between engagement in women’s group activities and the utilization of maternal and neonatal health services. This manuscript is well-written. Still, some major and minor comments are given below. I am happy to re-review it in the next round.
Major comments
=============
1. Study design. Lines 85–87: “The evaluation was done in these 26 intervention districts and 26 comparison districts that were selected to be similar to the intervention districts based on demographic and health services characteristics”.
Comparing outcomes (utilization of maternal and neonatal health services) between intervention and comparison population is interesting and worth reporting. Please explain the reason why the authors do not report this information?
2. Recall period.
Lines 88–90: “A baseline study was conducted before the intervention (December 2016 to February 2017) and an endline study two years after the baseline survey (December 2018 to February 2019)”.
Lines 122–123: “At least one antenatal care visits defines the proportion of pregnant women who received at least one antenatal care during pregnancy in the last 12 months preceding the survey”.
Please explain the reason why the authors restrict the recall period to 1 year period while the study design is for 2 years. This restriction also limited the sample size (only 896/6296) and reduced the power of statistical tests.
3. Choose outcome indicators. Other indicators for investigating RMNCH and progress toward UHC include at least four antenatal care visits, skilled birth attendance, institutional delivery, exclusive breastfeeding, and early initiation of breastfeeding (10.1016/j.lanwpc.2021.100230). Please explain the reasons for selecting those 3 reported indicators, and expand the indicators if possible.
4. Measure of economic status
Lines 133–134: “Economic status calculated based on household wealth which was categorized in one to five quintiles”.
Please explain how household wealth is evaluated, e.g., income-based, asset-based, or expenditure-based. Please present this in more detail.
5. Statistics test: In Table 2, the authors reported the use of the Chi-square test for P-values. However, as the sample size is limited, there are some cells with smaller-than-5 values. Please give discussions on the appropriateness of using the Chi-square test in this case and some potential solution if any.
Minor comments
=============
1. Introduction:
Some information in RMNCH service coverage of Ethiopia should be presented. Also, there should be a sentence discussion on the link between RMNCH service coverage and progress toward UHC (e.g., 10.1016/j.lanwpc.2021.100230).
2. Discussion:
The authors reported the coverage of some utilization of maternal and neonatal health services. They should make a comparison of their findings with national coverage. For example, the information could be found in Ethiopia DHS 2016 (https://dhsprogram.com/methodology/survey/survey-display-478.cfm)
Author Response
Response to Reviewer 3 Comments
Comments and Suggestions for Authors
Peer review:
IJERPH
Is women's engagement in Women’s Development Groups associated with enhanced utilization of maternal and neonatal health services? A cross-sectional study
General comments
=============
The manuscript described a cross-sectional study to evaluate the association between engagement in women’s group activities and the utilization of maternal and neonatal health services. This manuscript is well-written. Still, some major and minor comments are given below. I am happy to re-review it in the next round.
Response: Thank you for your encouraging words.
Major comments
=============
- Study design. Lines 85–87: “The evaluation was done in these 26 intervention districts and 26 comparison districts that were selected to be similar to the intervention districts based on demographic and health services characteristics”.
Comparing outcomes (utilization of maternal and neonatal health services) between intervention and comparison population is interesting and worth reporting. Please explain the reason why the authors do not report this information?
Response 1: Thank you. The intervention and comparison districts relate to the original study evaluation of the OHEP intervention that has been reported (Reference 14, Berhanu D, Okwaraji YB, Defar A, et al. Does a complex intervention targeting communities, health facilities and district health managers increase the utilisation of communitybased child health services? A before and after study in intervention and comparison areas of Ethiopia. BMJ Open 2020;10:e040868. doi:10.1136/ bmjopen-2020-040868) (Methods section, page 5, lines 136 to 138). The aim of the current secondary analysis is not related to the OHEP intervention.
- Recall period.
Lines 88–90: “A baseline study was conducted before the intervention (December 2016 to February 2017) and an endline study two years after the baseline survey (December 2018 to February 2019)”.
Lines 122–123: “At least one antenatal care visits defines the proportion of pregnant women who received at least one antenatal care during pregnancy in the last 12 months preceding the survey”.
Please explain the reason why the authors restrict the recall period to 1 year period while the study design is for 2 years.
This restriction also limited the sample size (only 896/6296) and reduced the power of statistical tests.
Response 2: Thank you. The information about the original evaluation study is provided to understand the context of this secondary analysis of data from the endline survey in the OHEP intervention. The aim of this secondary analysis is unrelated to the OHEP intervention. We included women who had delivered during the last 12 months to reduce recall bias (Methods section, on page 7, lines 180 and 181).
- Choose outcome indicators. Other indicators for investigating RMNCH and progress toward UHC include at least four antenatal care visits, skilled birth attendance, institutional delivery, exclusive breastfeeding, and early initiation of breastfeeding (10.1016/j.lanwpc.2021.100230). Please explain the reasons for selecting those 3 reported indicators, and expand the indicators if possible.
Response 3: Thank you. This information was available in the original dataset from the OHEP evaluation. Further, the Ethiopia Women Development Group activities manual mainly focuses on antenatal care, facility delivery and any postnatal care, which makes this restricted list reasonable (Reference 7: Ashebir F, M.A., Mulugeta A, Persson LÅ, Berhanu D Exploring women’s development group leaders’ support to maternal, neonatal and child health care: A qualitative study in Tigray region, Ethiopia. PLoS ONE, 2021 (16(9): e0257602).
- Measure of economic status
Lines 133–134: “Economic status calculated based on household wealth which was categorized in one to five quintiles”.
Please explain how household wealth is evaluated, e.g., income-based, asset-based, or expenditure-based. Please present this in more detail.
Response 4: Thank you. This household wealth index was based on assets and household characteristics. We have expanded the description of this index. (Methods section, pages 8, lines 223 to 226).
- Statistics test: In Table 2, the authors reported the use of the Chi-square test for P-values. However, as the sample size is limited, there are some cells with smaller-than-5 values.
Please give discussions on the appropriateness of using the Chi-square test in this case and some potential solution if any.
Response 5: Thank you. The intention of Table 2 is mainly to descriptively provide the distribtion of covariates in relation to utilization of services, not to test any associations. If you prefer us to delete the columns with p-values we could do so.
Further, we have recoded/ dichotomozed data WDG contact variable into “Neither WDG leader nor had contact” and “Had WDG contact not WDG leaders or WDG leaders”. The “Neither WDG leader nor had contact” represented as it is. The category “Had WDG contact not WDG leaders or WDG leaders”included women who were WDG leaders. Education variable also dichotomozed into “no education” and “Educated. The “no education” represented women who never had attended any formal education. The category “educated” included women with primary or secondary or higher level of edcuation. Birth order was also reclassified into one child, two or three children, and four or more children. (Methods section, on page 8, line 217 to 223). In regarding limited sample size, we have addressed these issues in the limitation part of the discussion section, page 17, lines 608 to 617.
Minor comments
=============
- Introduction:
Some information in RMNCH service coverage of Ethiopia should be presented. Also, there should be a sentence discussion on the link between RMNCH service coverage and progress toward UHC (e.g., 10.1016/j.lanwpc.2021.100230).
Response: Thank you. We have added more text in Introduction (Introduction section, on pages 3 & 4, lines 85 to 89) and in the discussion section, page 16, lines 593 to 596).
- Discussion:
The authors reported the coverage of some utilization of maternal and neonatal health services. They should make a comparison of their findings with national coverage. For example, the information could be found in Ethiopia DHS 2016 (https://dhsprogram.com/methodology/survey/survey-display-478.cfm)
Response: Thank you. We have made comparisions based on your comment (Discussion section, page 16, lines 593 to 596).
Round 2
Reviewer 2 Report
thank you very much for the effort, you have excellently taken up the improvements suggested by the reviewers.
Author Response
Response to Reviewer 2 to comment
Review Report round 2
Comments and Suggestions for Authors
Thank you very much for the effort, you have excellently taken up the improvements suggested by the reviewers.
Response: I thank you too much for your genuine support.

Reviewer 3 Report
Comments R2
Is women's engagement in Women’s Development Groups associated with enhanced utilization of maternal and neonatal health services? A cross-sectional study
General comments
=============
The manuscript described a cross-sectional study to evaluate the association between engagement in women’s group activities and the utilization of maternal and neonatal health services. This manuscript is well-written. Still, some major and minor comments are given below. I am happy to re-review it in the next round.
Response: Thank you for your encouraging words.
COMMENT: Thank you for your responses and revisions. Generally, I think the manuscript has been improved. Still, some major and minor comments should be addressed before the final decision is made.
Major comments
=============
1. Study design. Lines 85–87: “The evaluation was done in these 26 intervention districts and 26 comparison districts that were selected to be similar to the intervention districts based on demographic and health services characteristics”.
Comparing outcomes (utilization of maternal and neonatal health services) between intervention and comparison population is interesting and worth reporting. Please explain the reason why the authors do not report this information?
Response 1: Thank you. The intervention and comparison districts relate to the original study evaluation of the OHEP intervention that has been reported (Reference 14, Berhanu D, Okwaraji YB, Defar A, et al. Does a complex intervention targeting communities, health facilities and district health managers increase the utilisation of communitybased child health services? A before and after study in intervention and comparison areas of Ethiopia. BMJ Open 2020;10:e040868. doi:10.1136/ bmjopen-2020-040868) (Methods section, page 5, lines 136 to 138). The aim of the current secondary analysis is not related to the OHEP intervention.
COMMENT: Agree.
2. Recall period.
Lines 88–90: “A baseline study was conducted before the intervention (December 2016 to February 2017) and an endline study two years after the baseline survey (December 2018 to February 2019)”.
Lines 122–123: “At least one antenatal care visits defines the proportion of pregnant women who received at least one antenatal care during pregnancy in the last 12 months preceding the survey”.
Please explain the reason why the authors restrict the recall period to 1 year period while the study design is for 2 years.
This restriction also limited the sample size (only 896/6296) and reduced the power of statistical tests.
Response 2: Thank you. The information about the original evaluation study is provided to understand the context of this secondary analysis of data from the endline survey in the OHEP intervention. The aim of this secondary analysis is unrelated to the OHEP intervention. We included women who had delivered during the last 12 months to reduce recall bias (Methods section, on page 7, lines 180 and 181).
COMMENT: The reason of using 1-year recall period is not satisfied. Longer recall periods does not mean higher risk of recall bias. Specifically, Demographic and Health Survey (DHS) and Multiple Indicator Cluster Survey (MICS), the common data sources for RMNCH in LMICs, use longer recall periods in their surveys. More discussion in the choice of recall period should be added.
(https://www.who.int/data/gho/indicator-metadata-registry/imr-details/3323).
3. Choose outcome indicators. Other indicators for investigating RMNCH and progress toward UHC include at least four antenatal care visits, skilled birth attendance, institutional delivery, exclusive breastfeeding, and early initiation of breastfeeding (10.1016/j.lanwpc.2021.100230). Please explain the reasons for selecting those 3 reported indicators, and expand the indicators if possible.
Response 3: Thank you. This information was available in the original dataset from the OHEP evaluation. Further, the Ethiopia Women Development Group activities manual mainly focuses on antenatal care, facility delivery and any postnatal care, which makes this restricted list reasonable (Reference 7: Ashebir F, M.A., Mulugeta A, Persson LÅ, Berhanu D Exploring women’s development group leaders’ support to maternal, neonatal and child health care: A qualitative study in Tigray region, Ethiopia. PLoS ONE, 2021 (16(9): e0257602).
COMMENT: Agree.
4. Measure of economic status
Lines 133–134: “Economic status calculated based on household wealth which was categorized in one to five quintiles”.
Please explain how household wealth is evaluated, e.g., income-based, asset-based, or expenditure-based. Please present this in more detail.
Response 4: Thank you. This household wealth index was based on assets and household characteristics. We have expanded the description of this index. (Methods section, pages 8,
lines 223 to 226).
COMMENT: Agree.
5. Statistics test: In Table 2, the authors reported the use of the Chi-square test for
P-values. However, as the sample size is limited, there are some cells with smaller-than-5 values.
Please give discussions on the appropriateness of using the Chi-square test in this case and some potential solution if any.
Response 5: Thank you. The intention of Table 2 is mainly to descriptively provide the distribtion of covariates in relation to utilization of services, not to test any associations. If you prefer us to delete the columns with p-values we could do so.
Further, we have recoded/ dichotomozed data WDG contact variable into “Neither WDG leader nor had contact” and “Had WDG contact not WDG leaders or WDG leaders”. The “Neither WDG leader nor had contact” represented as it is. The category “Had WDG contact not WDG leaders or WDG leaders”included women who were WDG leaders. Education variable also dichotomozed into “no education” and “Educated. The “no education” represented women who never had attended any formal education. The category “educated” included women with primary or secondary or higher level of edcuation. Birth order was also reclassified into one child, two or three children, and four or more children. (Methods section, on page 8, line 217 to 223). In regarding limited sample size, we have addressed these issues in the limitation part of the discussion section, page 17, lines 608 to 617.
COMMENT: After revisions, some cells still contain smaller-than-5 values. I suggest the authors to perform Fisher's exact test for those variables.
Minor comments
=============
1. Introduction:
Some information in RMNCH service coverage of Ethiopia should be presented. Also, there should be a sentence discussion on the link between RMNCH service coverage and progress toward UHC (e.g., 10.1016/j.lanwpc.2021.100230).
Response: Thank you. We have added more text in Introduction (Introduction section, on pages 3 & 4, lines 85 to 89) and in the discussion section, page 16, lines 593 to 596).
COMMENT: I could not find the mentioned information. Specifically, it would be better if the authors could add a sentence emphasizing the importance of improving utilization of maternal and neonatal health services toward UHC in LMICs with proper reference(s) (10.1016/j.lanwpc.2021.100230)
2. Discussion:
The authors reported the coverage of some utilization of maternal and neonatal health services. They should make a comparison of their findings with national coverage. For example, the information could be found in Ethiopia DHS 2016 (https://dhsprogram.com/methodology/survey/survey-display-478.cfm)
Response: Thank you. We have made comparisions based on your comment (Discussion section, page 16, lines 593 to 596).
COMMENT: Agree.
Author Response
Response to Reviewer 2 for Comments
Comments and Suggestions for Authors
Is women's engagement in Women’s Development Groups associated with enhanced utilization of maternal and neonatal health services? A cross-sectional study
General comments
=============
The manuscript described a cross-sectional study to evaluate the association between engagement in women’s group activities and the utilization of maternal and neonatal health services. This manuscript is well-written. Still, some major and minor comments are given below. I am happy to re-review it in the next round.
Response: Thank you for your encouraging words.
COMMENT: Thank you for your responses and revisions. Generally, I think the manuscript has been improved. Still, some major and minor comments should be addressed before the final decision is made.
Major comments
=============
- Study design. Lines 85–87: “The evaluation was done in these 26 intervention districts and 26 comparison districts that were selected to be similar to the intervention districts based on demographic and health services characteristics”.
Comparing outcomes (utilization of maternal and neonatal health services) between intervention and comparison population is interesting and worth reporting. Please explain the reason why the authors do not report this information?
Response 1: Thank you. The intervention and comparison districts relate to the original study evaluation of the OHEP intervention that has been reported (Reference 14, Berhanu D, Okwaraji YB, Defar A, et al. Does a complex intervention targeting communities, health facilities and district health managers increase the utilisation of community based child health services? A before and after study in intervention and comparison areas of Ethiopia. BMJ Open 2020;10:e040868. doi:10.1136/ bmjopen-2020-040868) (Methods section, page 5, lines 136 to 138). The aim of the current secondary analysis is not related to the OHEP intervention.
COMMENT: Agree.
- Recall period.
Lines 88–90: “A baseline study was conducted before the intervention (December 2016 to February 2017) and an endline study two years after the baseline survey (December 2018 to February 2019)”.
Lines 122–123: “At least one antenatal care visits defines the proportion of pregnant women who received at least one antenatal care during pregnancy in the last 12 months preceding the survey”.
Please explain the reason why the authors restrict the recall period to 1 year period while the study design is for 2 years.
This restriction also limited the sample size (only 896/6296) and reduced the power of statistical tests.
Response 2: Thank you. The information about the original evaluation study is provided to understand the context of this secondary analysis of data from the endline survey in the OHEP intervention. The aim of this secondary analysis is unrelated to the OHEP intervention. We included women who had delivered during the last 12 months to reduce recall bias (Methods section, on page 7, lines 180 and 181).
COMMENT: The reason of using 1-year recall period is not satisfied. Longer recall periods does not mean higher risk of recall bias. Specifically, Demographic and Health Survey (DHS) and Multiple Indicator Cluster Survey (MICS), the common data sources for RMNCH in LMICs, use longer recall periods in their surveys. More discussion in the choice of recall period should be added.
(https://www.who.int/data/gho/indicator-metadata-registry/imr-details/3323).
Response 2: You are quite right. The decision to include 1-year recall period was primarily related to the original OHEP evaluation that included details regarding neonatal morbidity and treatment of illnesses. The 1-year period was chosen to reduce recall bias for such details. In addition, the majority of the respondents had illiterate. The current secondary analysis had therefore to be based on the 1-year recall data (in methods section, on page 7, lines 171 & 173). For example, References 8, 12, 17, 20, 21, 32 & 33 were analyzed using 1-year period before the survey.
- Choose outcome indicators. Other indicators for investigating RMNCH and progress toward UHC include at least four antenatal care visits, skilled birth attendance, institutional delivery, exclusive breastfeeding, and early initiation of breastfeeding (10.1016/j.lanwpc.2021.100230). Please explain the reasons for selecting those 3 reported indicators, and expand the indicators if possible.
Response 3: Thank you. This information was available in the original dataset from the OHEP evaluation. Further, the Ethiopia Women Development Group activities manual mainly focuses on antenatal care, facility delivery and any postnatal care, which makes this restricted list reasonable (Reference 7: Ashebir F, M.A., Mulugeta A, Persson LÅ, Berhanu D Exploring women’s development group leaders’ support to maternal, neonatal and child health care: A qualitative study in Tigray region, Ethiopia. PLoS ONE, 2021 (16(9): e0257602).
COMMENT: Agree.
- Measure of economic status
Lines 133–134: “Economic status calculated based on household wealth which was categorized in one to five quintiles”.
Please explain how household wealth is evaluated, e.g., income-based, asset-based, or expenditure-based. Please present this in more detail.
Response 4: Thank you. This household wealth index was based on assets and household characteristics. We have expanded the description of this index. (Methods section, pages 8, lines 223 to 226).
COMMENT: Agree.
- Statistics test: In Table 2, the authors reported the use of the Chi-square test for
P-values. However, as the sample size is limited, there are some cells with smaller-than-5 values.
Please give discussions on the appropriateness of using the Chi-square test in this case and some potential solution if any.
Response 5: Thank you. The intention of Table 2 is mainly to descriptively provide the distrbution of covariates in relation to utilization of services, not to test any associations. If you prefer us to delete the columns with p-values we could do so.
Further, we have recoded/ dichotomized data WDG contact variable into “Neither WDG leader nor had contact” and “Had WDG contact not WDG leaders or WDG leaders”. The “Neither WDG leader nor had contact” represented as it is. The category “Had WDG contact not WDG leaders or WDG leaders”included women who were WDG leaders. Education variable also dichotomized into “no education” and “Educated. The “no education” represented women who never had attended any formal education. The category “educated” included women with primary or secondary or higher level of education. Birth order was also reclassified into one child, two or three children, and four or more children. (Methods section, on page 8, line 217 to 223). In regarding limited sample size, we have addressed these issues in the limitation part of the discussion section, page 17, lines 608 to 617.
COMMENT: After revisions, some cells still contain smaller-than-5 values. I suggest the authors to perform Fisher's exact test for those variables.
Response: It is well accepted. We have corrected as your suggestion it in Methods section, on page 9, lines 247 and 248; and on Results section on Table 2, page 13, line 304; and line 306.
Minor comments
=============
- Introduction:
Some information in RMNCH service coverage of Ethiopia should be presented. Also, there should be a sentence discussion on the link between RMNCH service coverage and progress toward UHC (e.g., 10.1016/j.lanwpc.2021.100230).
Response 1: Thank you. We have added more text in Introduction (Introduction section, on pages 3 & 4, lines 85 to 89) and in the discussion section, page 16, lines 593 to 596).
COMMENT: I could not find the mentioned information. Specifically, it would be better if the authors could add a sentence emphasizing the importance of improving utilization of maternal and neonatal health services toward UHC in LMICs with proper reference(s) (10.1016/j.lanwpc.2021.100230)
Response 1: Thank you. We have added in Introduction section, page 3, lines 74 to 80)
- Discussion:
The authors reported the coverage of some utilization of maternal and neonatal health services. They should make a comparison of their findings with national coverage. For example, the information could be found in Ethiopia DHS 2016 (https://dhsprogram.com/methodology/survey/survey-display-478.cfm)
Response 2: Thank you. We have made comparisons based on your comment (Discussion section, page 16, lines 593 to 596).
COMMENT: Agree.
